# In Vitro Antifungal Antibacterial Activity of Partitions from *Euphorbia tirucalli* L.

Michel Stéphane Heya [1], María Julia Verde-Star [2], Sergio Arturo Galindo-Rodríguez [2], Catalina Rivas-Morales [2], Efrén Robledo-Leal [3] and David Gilberto García-Hernández [2],*

[1] Institute of Biotechnology, Facultad de Ciencias Biológicas, Universidad Autónoma de Nuevo León, San Nicolás de los Garza 66455, Nuevo León, Mexico; heyamichelstephane@yahoo.fr

[2] Laboratory of Phytochemistry, Department of Chemistry, Facultad de Ciencias Biológicas, Universidad Autónoma de Nuevo León, San Nicolás de los Garza 66455, Nuevo León, Mexico; jverdestar@gmail.com (M.J.V.-S.); sagrod@yahoo.com.mx (S.A.G.-R.); catalinarivas@yahoo.com.mx (C.R.-M.)

[3] Laboratory of Mycology, Department of Microbiology and Immunology, Facultad de Ciencias Biológicas, Universidad Autónoma de Nuevo León, San Nicolás de los Garza 66455, Nuevo León, Mexico; efren.robledoll@uanl.edu.mx

* Correspondence: david.garciahrz@uanl.edu.mx

**Abstract:** We determined the antifungal and antimicrobial sensitivity of *Euphorbia tirucalli* extracts in vitro. Antifungal and antibacterial activity was determined based on the M38-A and M26-A protocols, respectively. The methanolic and ethanolic partitions demonstrated antidermatophytic activity against *Trichophyton rubrum* (MIC 125 μg/mL for ethanol and MIC 125 μg/mL for methanol) and *T. interdigitalis* (MIC 500 μg/mL for ethanol; 125 μg/mL for methanol). These partitions also showed antibacterial activity—the ethanolic partition had an MIC of $1.56 \pm 0.02$ mg/mL against methicillin-resistant *Staphylococcus aureus* (clinical isolate), $6.25 \pm 0.04$ mg/mL against *Staphylococcus aureus* BAA-44, $3.13 \pm 0.13$ mg/mL against *Pseudomonas aeruginosa* 27853, and $3.13 \pm 0.15$ mg/mL against *Escherichia coli* ATCC 25922; the methanolic partition showed an MIC of $1.56 \pm 0.02$ mg/mL against *P. aeruginosa* 27853 and $1.56 \pm 0.043$ mg/mL against *E. coli* ATCC 25922. These partitions show promise as antimicrobial agents or adjuvants in the treatment of infections caused by these microorganisms.

**Keywords:** antibiotic; clinical isolates; dermatophytes; antidermatophytic activity; multidrug-resistant

## 1. Introduction

In the last three decades the pharmaceutical industry has seen a renewed interest in natural products and their possible applications in the search for new and more efficient drugs. Plants are an important source for drug production; in effect, numerous studies have confirmed the biological potential of phytochemical compounds against human diseases, with antimicrobial [1,2], antiviral [3], antioxidant [2,4], and antidiabetic properties [5], among others. It is worth mentioning that about 30% of the drugs used in industrialized countries have been synthesized from plant products [6]. Within the wide range of plants with pharmacological potential, the *Euphorbia* genus is among the most used in traditional medicine in many parts of the world, and reports of its antimicrobial activity has aroused interest in the scientific community [7]. *Euphorbia tirucalli* is a native plant of southern Africa, widely used in Indian and Brazilian ethnomedicine in the treatment of skin conditions such as excrescences, nodules, abscesses, warts, epithelioma, sarcoma, skin tumors, and leprosy, among others [8]. The alcoholic extracts of the stem bark and leaves of *E. tirucalli* have antimicrobial activity against *P. aeruginosa, Candida tropicalis, C. albicans*, and *Aspergillus niger*. Stem extracts with acetone, hexane, methanol, chloroform, and petroleum are reported to have antibacterial activity against *E. coli* and *Bacillus megaterium* [9], whereas the aqueous extract of aerial parts eliminates superoxide anions and scavenges hydroxyl radicals [10]. Based on these properties, it is necessary to continue researching the pharmacological properties of *E. tirucalli*.

The number of human diseases caused by microorganisms has increased, which is mainly attributed to a deficiency in immune function, along with the low bioavailability of antifungals and antibiotics [11]. According to the WHO, fungal mycoses have a global prevalence of 20% to 25% for the general population; 5% to 10% of these cases are caused by dermatophytes, with *Trichophyton rubrum* and *T. interdigitalis* being the most frequent species. The incidence of chronic dermatophytosis has sparked interest in seeking alternative solutions to combat these diseases [12–14]. Another pressing issue is presented by nosocomial infections caused by organisms acquired during stays in health centers, which are an important cause of morbidity and mortality in hospitals and incur an increase in medication costs. Global prevalence is relatively high, reaching 10% of the population; more people die each year from nosocomial diseases caused by multidrug-resistant bacteria than from homicides and traffic accidents [15]. These infections are commonly caused by *Escherichia coli, Staphylococcus aureus*, and *Pseudomonas aeruginosa*, which cause conditions in the urinary and digestive tracts (diarrhea), toxic shock syndrome, pneumonia, and septicemia [16–18].

Due to the continued use of medicinal plants, it is necessary to continue using valid experimental models to demonstrate their therapeutic properties. We evaluated the in vitro antifungal activity of different partitions of extracts obtained from *E. tirucalli* against clinical isolates of dermatophytes (*T. rubrum* and *T. mentagrophytes*), as well as against bacterial strains responsible for nosocomial infections (*S. aureus*, *P. aeruginosa*, and *E. coli*).

## 2. Materials and Methods

### 2.1. Microorganisms

Isolates of the fungi *Trichophyton rubrum* and *T. interdigitalis*, as well as the bacterial strains *Escherichia coli* 25922, *Staphylococcus aureus* BAA-44, *Staphylococcus aureus* clinical isolate, and *Pseudomonas aeruginosa* 27853, were used. All strains were obtained from the microorganism collection from the Laboratory of Analytical Chemistry of the College of Biological Sciences of the Universidad Autónoma de Nuevo León (UANL).

### 2.2. Collection of Plant Material

The *E. tirucalli* (E.t) collection was carried out in northern San Nicolás de los Garza, in Nuevo León, México. The samples were taxonomically classified in the Botany Department of the College of Biological Sciences in UANL, and were registered in the herbarium under folio number: 029755. The plant material was repeatedly washed with distilled water and dried at 40 °C with a white light lamp (150 W). Subsequently, the material was mashed using a manual grinder.

### 2.3. Preparation of Extracts

First, 40 g of dried and ground plant material was subjected to continuous extraction using Soxhlet equipment with hexane, chloroform, ethanol, and methanol (CTR Scientifics). Partitions were recovered and solvents were removed under reduced pressure (Yamato rotary evaporator model RE200). The plant material was brought to dryness in a stove oven at a temperature no higher than 40 °C (BTC-9100, TERLAB, Zapopan, México). Once solvent-free, all four partitions (hexane: E.t-Hex-Part; chloroform: E.t-Clo-Part; ethanol: E.t-EtOH-Part; methanol: E.t-MeOH-Part) were stored at ambient temperature until use (Figure 1).

The yield was obtained by the rule of three, considering the initial plant material (40 g) as 100%.

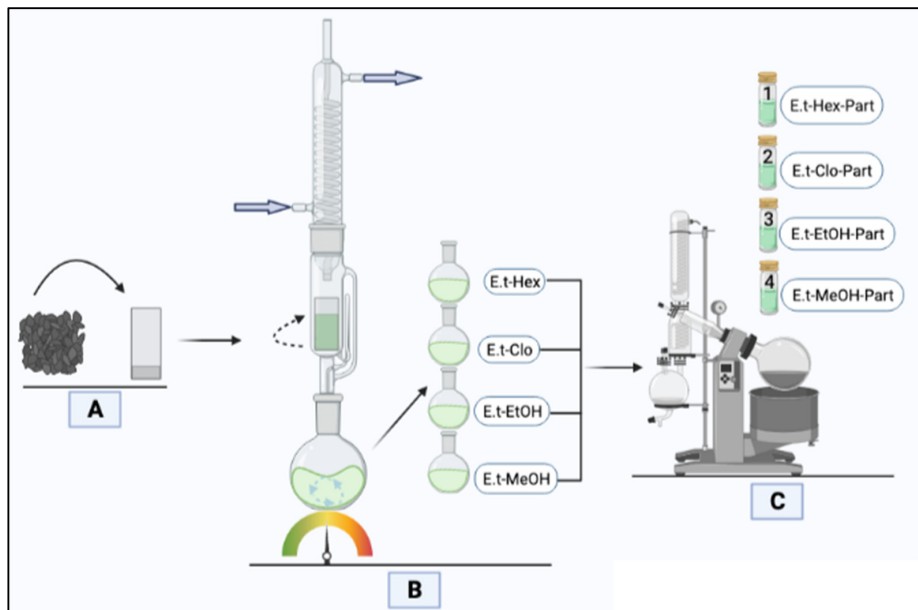

**Figure 1.** Flowchart of obtaining the partitions of E. tirucalli: (**A**) Filling the extractor by placing the dry plant material in a filter jacket. (**B**) Successive and independent distillations of the vegetal framework with different solvents (hexane, chloroform, ethanol, and methanol) to obtain the partitions. (**C**) Evaporation, under reduced pressure, of each partition to recover them free of solvents (1 < 2 < 3 < 4).

### 2.4. Preparation of Treatments

Plant treatment stocks (20 mg/mL with corresponding solvent) were prepared and homogenized by ultrasound (Ney ultrasonic cleaner 19 h) for 20 min. For antibacterial activity, initial solutions of 25 mg/mL were prepared via the same procedure.

### 2.5. Preparation of Inoculate

*For the dermatophyte strains:* Strains were initially activated on potato dextrose agar (PDA; CTR Scientifics, Monterrey, México), for 21 days at 28 °C (Quincy lab, Model 12–140 Bench Top Incubator, Waltham, MA, USA). Subsequently, fungal colonies were covered with 10 mL of sterile distilled water, and their surfaces were gently scraped with a sterile loop. Later, the obtained mixtures were filtered with a sterile gauze to separate the conidia from the hyphae and agar particles. Before performing biological tests, conidia suspensions were adjusted to $1–3 \times 10^3$ CFU/mL concentrations after being counted with a hemocytometer (CTR Scientifics, Monterrey, México).

*For the bacterial strains*: Strains were inoculated in Müller–Hinton (CTR Scientifics, Monterrey, Nuevo León, México) broth and incubated at 24 h at 35 °C. Biological tests were performed after adjusting the bacterial solution to a concentration of $1 \times 10^8$ CFU/mL (Matiz Melo et al., 2015) [19].

### 2.6. Determination of the Antidermatophytic Activity of the E. tirucalli L. Partitions

In vitro antifungal sensitivity was determined through modification of the microdilution method described in the M38-A protocol by the Clinical for Laboratory Standards Institute, 2008 [20], using clotrimazole (Sigma-Aldrich, St. Louis, MI, USA) as a control. Partition solutions were diluted from 20,000 µg/mL to 8000 µg/mL to carry out subsequent dilutions. With respect to the positive controls, 1000 mg/mL dilutions were obtained from stock solutions (1000 µg/mL).

Initially, 100 mL of MH (Müller–Hinton) medium was prepared, and 0.02 mg of phenol red was added before sterilization (MH-Ph). Subsequently, the biological testing was carried out through serial dilutions in flat-bottomed 96-well microplates, using MH-Ph broth as a

diluent from an initial partition concentration of 8000 µg/mL. To the MH-Ph-Part broth initially placed in the microplate, 100 mL of conidia solution was added in order to obtain final concentrations of 0.0076–2000 µg/mL. With respect to the control with clotrimazole (CLSI, 2008), final concentrations of 0.0001–25 µg/mL were obtained. Dilutions without inoculum were used as blanks; methanol dilutions were used as negative controls. For growth control, 100 µL of each strain was inoculated in 100 µL of medium. The obtained mixtures were incubated at 28 °C for 120 h (Figure 2).

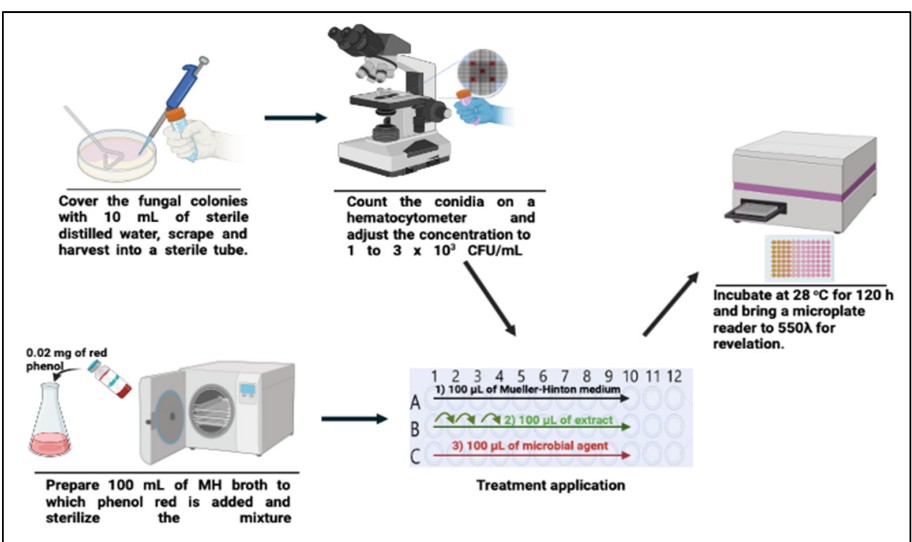

**Figure 2.** Flowchart to determine the antifungal activity.

Antifungal activity was determined using the minimum inhibitory concentration (MIC), according to the CLSI as the concentration capable of inhibiting the fungal growth (% H) at 80% compared to the control growth. Revelation was determined with a microplate ELx800 universal microtiter plate reader (BioTek, Santa Clara, CA, USA) at 550 λ, and the inhibition percentage (% H) was determined using the following formula

$$\%H = [Control(Abs) - (Treatment(Abs) - blank(Abs))/Control(Abs)] \times 100$$

### 2.7. Antibacterial Activity of E. tirucalli L. Partitions

MIC values for plant extracts were determined according to the M26-A protocol (CLSI, 1999) [21]. The inocula were prepared from 18 h broth cultures, and suspensions were adjusted to 0.5 McFarland standard turbidity. Partitions were first diluted to the highest concentration, and then serial 2-fold dilutions were carried out in a concentration range from 25–0.781 mg/mL (final concentrations: 12.5–0.391 mg/mL). MIC values against bacteria were determined based on a micro-well dilution method. Microbial growth was determined by absorbance values at 540 nm using an ELx800 universal microtiter plate reader (BioTek, Santa Clara, CA, USA). The MIC was defined as the lowest compound concentration to inhibit microorganism growth. In order to reveal whether a partition had bacteriostatic or bactericidal activity, a sample was removed from the MIC and inoculated in Müller–Hinton agar. Growth in the medium was considered bacteriostatic, whereas no growth was considered bactericidal. All assays were carried out in triplicate.

## 3. Results

### 3.1. Percentage of Extraction Yield

Successive extractions were carried out from 40 g of dry plant material. The partition yield for hexane was 4.63%; for chloroform, 3.028%; for ethanol, 7.775%, and for methanol, 12.423% (Table 1).

**Table 1.** Percentages of extraction yield, efficacy, and performance, after 48 h of extraction.

| Solvents | IQ (g) | OQ (g) | R (%) | Efficacy (%) | Performance (%) |
|---|---|---|---|---|---|
| Hexane | 40 | 1.852 | 4.630 | 4.630 | 4.630 |
| Chloroform | 38.789 | 1.211 | 3.028 | 3.122 | 3.122 |
| Ethanol | 37.578 | 3.110 | 7.775 | 8.276 | 8.276 |
| Methanol | 34.468 | 4.969 | 12.423 | 14.416 | 14.416 |

IQ: quantity of plant material at each stage; OQ: quantity of obtained extract; R: yield. The yield was calculated from the initial amount of plant material (40 g).

## 3.2. Antifungal Activity

The methanolic and ethanolic partitions had inhibitory effects on the growth of clinical dermatophyte isolates. The two dermatophytic strains had an MIC of 125 µg/mL for the methanolic partition, whereas for ethanolic partitions the MIC for *T. rubrum* was 125 µg/mL, while for *T. interdigitalis* it was 500 µg/mL (Table 2). No inhibitory effect was observed on dermatophyte growth in solvent controls (methanol and ethanol); therefore, the solvents do not contribute to the partitions' activity. Moreover, using a blank in the formula to evaluate the inhibition eliminated the absorbance effect of the extract.

**Table 2.** Fungal activity of the methanolic and ethanolic partitions of *E. tirucalli* against clinical isolates of dermatophytes.

| Dermatophytic Strains | E.t-MeOH-Part | | E.t-EtOH-Part | | Clotrimazole | |
|---|---|---|---|---|---|---|
| | % H | MIC (µg/mL) | % H | MIC (µg/mL) | % H | MIC (µg/mL) |
| *T. rubrum* | 80.25 ± 2.06 | 125 | 81.11 ± 2 | 125 | 82.30 ± 0.94 | 0.1 |
| *T. interdigitalis* | 86.12 ± 4.08 | 125 | 81.99 ± 3.53 | 500 | 92.57 ± 3.11 | 0.0004 |

$n$ = 3. Results in µg/mL—E.t-MeOH-Part: methanolic partition of *E. tirucalli*; E.t-EtOH-Part: ethanolic partition of *E. tirucalli*; % H: inhibition percentage.

The same MICs were obtained in both the ethanolic and methanolic partitions against *T. rubrum*, whereas a highly significant difference was found between the MICs of *T. rubrum* and *T. mentagrophytes* (Table 2). Qualitatively, changes in the color of culture medium to fuchsia red indicated the growth of fungus, whereas slight changes in color showed a low inhibition of fungal growth.

## 3.3. Antibacterial Activity

Both partitions showed growth inhibition of both Gram-positive and -negative strains, but only the ethanolic partition had activity against MRSA strains (ATCC and Clinical Isolates). For the methanolic partition, the bacteriostatic activity is shown in Table 3. The ethanolic partitions demonstrated bactericidal activity against *P. aeruginosa* ATCC 27853 (3.125 ± 0.129) and *S. aureus* BAA-44, (6.25 ± 0.038 mg/mL). These results are interesting because these bacteria are usually common etiological agent for nosocomial diseases.

**Table 3.** Determination of the minimum inhibitory concentrations (MICs) of the alcoholic partitions of *E. tirucalli*, and the corresponding types of biological activity.

| Clinically Important Strain | Partition (Part.) | MIC (mg/mL) | Biological Activity |
|---|---|---|---|
| *P. aeruginosa* ATCC 27853 | E.t-MeOH | 1.56 ± 0.02 | Bacteriostatic |
| *P. aeruginosa* ATCC 27853 | E.t-EtOH | 3.13 ± 0.13 | Bactericidal |
| *E. coli* ATCC 25922 | E.t-MeOH | 1.56 ± 0.04 | Bacteriostatic |
| *E. coli* ATCC 25922 | E.t-EtOH | 3.13 ± 0.15 | Bacteriostatic |
| *S. aureus* BAA-44 | E.t-EtOH | 6.25 ± 0.04 | Bactericidal |
| *S. aureus* clinical isolate | E.t-EtOH | 1.56 ± 0.02 | Bacteriostatic |

$n$ = 3; E.t-MeOH: methanolic partition; E.t-EtOH: ethanolic partition.

## 4. Discussion

The methanolic and ethanolic partitions obtained in this study showed both bactericidal and bacteriostatic activity against antibiotic-resistant microorganisms; the ethanolic partition showed a greater bactericidal potential against *P. aeruginosa* ATCC 27853 (3.125 mg/mL). This compares to the agar plating method used by Kumar et al. 2010 [10], who found that a concentration of 2.5 mg/mL inhibited approximately 40% of growth. For *S. aureus,* our methods showed 100% inhibition, whereas Kumar et al. (2010) [10] showed an inhibition of 30.76% at a concentration of 2.5 mg/mL compared to its inhibition control (streptomycin). It should be noted that the strain used in our work is considered multidrug-resistant, and did not present any resistance; however, at 6.25 mg/mL it already had a 100% inhibition, which translates into a bactericidal effect. In relation to the methanolic partitions, our results show greater inhibitory potential at concentrations below 2.5 mg/mL. The differences between studies suggest that when partitioning from nonpolar solvents to methanol, the metabolites present behave either as bacteriostatic or bactericidal, because they are separated by their polarity gradients.

The in vitro sensitivity of partitions obtained from *E. tirucalli* against clinical isolates of dermatophytes and multidrug-resistant bacteria were determined using clotrimazole as a control. The E.t-MeOH-Part and E.t-EtOH-Part were found to have antifungal activity against dermatophyte clinical isolates (*T. rubrum* and *T. interdigitalis*) and multidrug-resistant bacterial strains; *T. rubrum* showed an MIC of 125 μg/mL with the E.t-EtOH-Part, while *T. interdigitalis* had an MIC of 500 μg/mL. Both dermatophytic strains had an MIC of 125 μg/mL with the E.t-MeOH-Part (Table 2). It should be noted that these results are similar to those obtained by Kumar et al. in 2010, who introduced a methanolic extract of *E. tirucalli* to strains of clinical importance such as *Candida albicans, C. tropicalis, Aspergillus niger, A. fumigatus, A. flavus*, and *Fusarium oxysporum* [10]. Our MICs obtained with E.t-MeOH-Part were similar to those obtained by Parekh and Chanda (2008) for different strains of *Candida* [22]. However, the slight improvement in MIC that we found could be related to the partitions obtained, which could favor the solubility of secondary metabolites, thereby increasing the contact surface on microorganisms. According to Savjani et al. (2012), the phenomenon of creating dilutions from a stock solution with the purpose of obtaining a homogeneous system is one of the important parameters used to achieve the desired concentration for a strong pharmacological response [23]. Furthermore, secondary metabolites such as tannins and other phenolic compounds are classified as highly active antimicrobial compounds [24]. Extract partitions from *E. tirucalli* L. revealed the presence of tannins, flavonoids, triterpenes, and sterols, which are biologically active against various microbial agents responsible for human diseases [24–26].

Although the two partitions had different minimum fungicide concentrations, they presented equal MICs (125 μg/mL) against *T. rubrum* (Table 2). This is related to the small differences in their phytochemical profiles, given that the E.t-EtOH-Part showed the presence of tannins, flavonoids, triterpenes, and sterols, whereas the phytochemical profile of the E.t-MeOH-Part exhibited the presence of tannins, flavonoids, sterols, and sesquiterpenes, in addition to those previously mentioned. According to the results obtained by Miron et al. (2014) [27], the antifungal activity of these partitions may be related to the presence of terpenes. Dermatophytes (e.g., *T. rubrum, T. mentagrophytes, Microspurum gypseum*, and *M. canis*) are considerably sensible to terpenes (i.e., monoterpenes). Qasim and Rasool [28,29] found that the terpenes taraxerano and cycloeuphordenol are present in the aerial parts of *E. tirucalli* L. Because the main difference between the E.t-EtOH-Part and the E.t-MeOH-Part is the presence of sesquiterpenes in the latter, the sensitivity shown by *T. interdigitalis* may be related to this compound. Duong et al. (2019) reported the presence of sesquiterpenes in *E. tirucalli* L.; these compounds have shown antibiotic activity against different organisms, and this may be the case for *T. interdigitalis* [30]. In a previous study, the methanolic partition of *E. tirucalli* L. showed antimicrobial activity against *B. subtilis*, *E. coli*, *E. faecalis*, and *C. albicans* [31]. Partitions obtained from *E. tirucalli* L. also contain flavonoids and tannins—compounds generally recognized for their antioxidant and antimi-

crobial potential, as well as their role in collagen synthesis [32,33]. It is possible that synergy against dermatophytic strains occurs between the compounds present in the partitions [34].

## 5. Conclusions

In the present study, ethanolic and methanolic partitions of *E. tirucalli* L. showed antimicrobial potential against clinical isolates of dermatophytes (*T. rubrum* and *T. interdigitalis*) and bacteria (*E. coli* 25922, *S. aureus* BAA44, *S. aureus* MRSA, and *P. aeruginosa* 27853). We found that the ethanolic partition had the highest biological activity against bacteria, whereas the methanolic partition was most effective against dermatophytes. These biological activities were related to the phytochemical profiles of each of the partitions. The secondary metabolites of *E. tirucalli* are promising antimicrobial agents that could be used in the treatment of the diseases caused by these microorganisms. Although the pharmacological potential of these partitions was evidenced, it is worth mentioning that even though Soxhlet extraction offers numerous advantages, the use of high temperatures in this method increases the possibilities of thermal degradation of compounds such as polyphenols and alkaloids, generally known for their antimicrobial potential. Therefore, in the future we will try to evaluate the same biological tests using partitions of *E. tirucalli* obtained via other extraction methods that use more tolerable temperatures.

**Author Contributions:** Conceptualization, M.S.H. and D.G.G.-H.; Methodology, M.S.H. and D.G.G.-H.; Software, M.S.H.; Validation, M.J.V.-S., S.A.G.-R. and C.R.-M.; Formal Analysis, M.S.H., D.G.G.-H. and S.A.G.-R.; Investigation, M.S.H. and D.G.G.-H.; Resources, M.J.V.-S and C.R.-M.; Data Curation, M.S.H., D.G.G.-H., and S.A.G.-R.; Writing—Original Draft Preparation, M.S.H. and D.G.G.-H.; Writing—Review and Editing, all authors; Visualization, E.R.-L.; Supervision, D.G.G.-H., S.A.G.-R., C.R.-M. and E.R.-L.; Project Administration, M.J.V.-S., D.G.G.-H. and C.R.-M.; Funding Acquisition, M.J.V.-S. All authors have read and agreed to the published version of the manuscript.

**Funding:** This research received no external funding.

**Institutional Review Board Statement:** Not applicable.

**Informed Consent Statement:** Not applicable.

**Data Availability Statement:** Not applicable.

**Acknowledgments:** We give thanks to the National Council of Science and Technology for the scholarship (CVU: 623209), and the Alejandra Elizabeth Arreola-Triana for her observations and comments.

**Conflicts of Interest:** The authors declare no conflict of interest.

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
