# Peer review of "In Vitro Antifungal Antibacterial Activity of Partitions from Euphorbia tirucalli L."

_analytica, doi:10.3390/analytica3020016_

Round 1

Author Response

Response to Reviewer 1 Comments

Point 1: I The topic of manuscript is current, the introduction is clear, however determination antimicrobial activity was performed at too low temperature (28°C). According to NCCLS guidelines and reference included in the reviewed manuscript Matiz Melo et al. 2015 the temperature should be 35°C± 2. Moreover, novelty elements in the extraction procedure should be indicated.

Response 1: It was an editing error; the value was corrected.

Point 2: In my opinion we do not have evidence that ” the ethanolic partition showed a greater bactericidal potential against P. aeruginosa ATCC 27853 (3.125 214 mg/mL)” -lines 213-214, because bactericidal potential of methanolic partition was only investigated at 1.56 ± 0.02mg/mL. Bactericidal potential of methanolic partition should be also check at 3.125 ± 0.129mg/mL.

Response 2: In the work, it is mentioned that the antibacterial activity was determined in a concentration range of 25 to 0.781 mg/mL; however, the methanolic partition has been bacteriostatic at 1.56 ± 0.02mg/mL (table 3) at higher concentrations. On the other hand, the ethanolic partition presented a bactericidal potential (3.125 ± 0.129mg/mL), which makes it the most effective partition.

Point 3: References are not always written in accordance with guidelines. Sometimes titles of the journals are missing for instance: 19, 22, 25, 28. The way of writing page numbers is different.

Response 3: These references have been written correctly.

Point 4: Lines 66-67, It is written two times P. aeruginosa.

Response 4: Duplication was deleted

Point 5: Line 84, it should be 40 g instead of “40 gr”

Response 5: gr has been changed to g

Point 6: In Tables 1 and 2 the numbers of lines joined with tables.

Response 6: The tables have been inserted correctly in the text.

Point 7: There are not explanations for values included in the brackets in Table 2.

Response 7: The table has been reorganized for better understanding.

Point 8: Line 274, It should be dermatophytes instead of “darmatophytes”

Response 8: The word has been written correctly.

Reviewer 2 Report

  1. More background introduction related to the use of natural antibiotics to treat infections should be added ;
  2. Knowledge gaps concerning extraction and utilization of plant antibiotics should be presented
  3. Statistical analyses regarding the performance of extraction and efficacy should be added.

Author Response

Response to Reviewer 2 Comments

Point 1: I More background introduction related to the use of natural antibiotics to treat infections should be added.

Response 1: More background on the biological potential of plant extracts has.

Point 2: Knowledge gaps concerning extraction and utilization of plant antibiotics should be presented.

Response 2: This specification was added to the conclusion, along with prospects.

Point 3: Statistical analyses regarding the performance of extraction and efficacy should be added.

Response 3: Statistical data was added to table 1.

Round 2

Author Response

Point 1: „the methanolic partition has been bacteriostatic at 1.56 ± 0.02mg/mL (table 3) at higher concentrations”
This sentence is not clear. In the manuscript should be included information that the methanolic partition has been bacteriostatic at 1.56 ± 0.02mg/mL (table 3) and at higher concentrations.

Response 1: In the methodology, it has been defined how the conclusion of the bactericidal and bacreriostatic potential of the treatments was reached.

Point 2: Determination antifungal activity was performed using Mueller Hinton broth. This medium is recommended by FDA, World Health Organization and NCCLS for testing most commonly encountered aerobic and facultative anaerobic bacteria in food and clinical material. Usually optimal growth of fungi is supported by RPMI 1640 and Sabouraud dextrose broth. The use of MH broth should be justified.

Response 2: It is true that the FDA, the World Health Organization and the NCCLS report the use of RPMI 1640 and Sabouraud broth as nutrient media to determine antidermatophytic susceptibility in vitro, with a preculture of the fungal strains in PDA medium to allow good growth. of the conidia. However, within the reliable methods proposed for Antifungal Susceptibility Testing (AST) to determine antidermatophytic activity in vitro, the Mueller Hinton agar plate diffusion technique described by Nweze et al., in 2010 [1, 2]; indicating that dermatophytes can also grow extensively on Mueller-Hinton medium. Several scientific studies have relied on the proposed NCCLS protocol with certain modifications to determine the multidrug susceptibility of dermatophytic agents in vitro [3, 4].

Point 3: Page 6, line 227, it should be inhibition instead of “Inhibition”

Response 3: The word has been written correctly.

Point 4: References are not always written in accordance with guidelines. The whole list should be read again because contains editing errors. The way of writing page numbers is different (for instance in reference 19 and 28 pages are given in two different ways), in reference 34 should be included only abbreviation of the journal title, in some references spaces are missing.

Response 4: All references have been written correctly.

Note: In the new manuscript, we have greatly improved English.

References:

  1. Thatai, P.; Sapra, B. Critical review on retrospective and prospective changes in antifungal susceptibility testing for dermatophytes. Mycoses. 2016, 59, 615–627. doi: 10.1111/myc.12514.
  2. Nweze, E.I.; Mukherjee, P.K.; Ghannoum, M.A. Agar-based disk diffusion assay for susceptibility testing of dermatophytes. Clin. Microbiol. 2010, 48, 10,v3750–3752. doi: 10.1128/JCM.01357-10.
  3. Morales, J.L.; Cantillo-Ciau, Z.O.; Sánchez-Molina, I.; Mena-Rejón, G.J. Screening of antibacterial and antifungal activities of six marine macroalgae from coasts of Yucatán peninsula. Biol. 2006, 44, 632–635. doi: 10.1080/13880200600897569.
  4. Abd El Alim, M.; Abdel Halim, R.M.; Habib, S.A. Comparison of Broth Micro Dilution and Disk Diffusion Methods for Susceptibility Testing of Dermatophytes. J. Hosp. Med. 2017, 69, 1923–1930. doi: 10.12816/0040624.
